# Temporal and altitudinal variability of the Spread F observed by the VHF radar over Christmas Island

Ricardo Yvan de La Cruz Cueva[1], Eurico Rodrigues de Paula[2], Acácio Cunha Neto[2]

[1]Physics Department, State University of Maranhão, São Luís, Maranhão, Brazil.
[2]DIHPA- Heliophysics, Planetary Sciences and Aeronomy Division, National Institute for Space Research, São José dos Campos, São Paulo, Brazil.

*Correspondence to*: Ricardo Y. C. Cueva (navivacu@gmail.com)

**Abstract.** The goal of this work is to study the time and altitude echoes characteristics under different solar and seasonality conditions using the VHF radar RTI images. The occurrence of equatorial spread F depends on the existence of conditions that can seed the Rayleigh-Taylor instability, and these conditions can change with solar flux, seasonality, longitude distributions, and day-to-day variability. So, the equatorial spread F is observed as its time and altitude occurrence. The VHF

radar of Christmas Island ($2.0^{o}$ N, $157.4^{o}$ W, 2.9ºN dip latitude) has been operational in the equatorial region for some time, allowing long-term observations. The occurrence of echoes during solar minimum conditions is observed throughout the night since the post reversal westward electric field is weaker than the solar maximum and the possibilities for the vertical plasma drift to become positive are larger. On other hand, echoes during solar maximum will be controlled by dynamics near the time of the Pre-reversal Peak (PRE). Our results indicate that the peak time occurrence of echoes along this period shows

a well-defined pattern, with echoes distributed as closer to local sunset during solar maximum and around/closer to midnight during solar minimum conditions, meanwhile, the peak altitude occurrence of echoes shows a slightly regular pattern with higher altitude occurrences during solar maxima and lower altitudes during solar minimum conditions.



## 1 Introduction

The contemporaneous understanding of the formation of F-region plasma irregularities depends mainly on the Rayleigh-Taylor (RT) instability process, due to its appearance at the bottomside of the F-region, then becoming unstable to finally generate plasma bubbles. These newly formed plasma bubbles evolve in a nonlinear process and then extend into high altitudes into the F-region. The small-scale (centimeter to a few tens of meters) irregularities formed in this process are responsible for radar backscatter, which structures in the range-time-intensity (RTI) image of the radar. The pioneering

ionospheric radar work of Woodman and LaHoz (1976) attributed the term "plumes" to describe radar echoes reaching the topside ionosphere. They observed a slope in the formation of the plumes, then explained using numerical simulation by Ossakow (1981) and Zalesak et al. (1982).

The RT instability (and ESF) is controlled by a number of parameters like the prereversal enhancement (PRE) of the zonal

equatorial electric field, zonal and meridional neutral winds, longitudinal conductivity gradients, flux tube integrated conductivities, and, possibly, variations in initial (or seed) perturbations (Abdu, 2001; Fejer et al., 1999). It has been noted that ESF bubbles at pre-midnight and post-midnight hours could be driven by different mechanisms (Dao et al., 2011; Yizengaw et al., 2013). The mechanisms that should control the appearance or suppression of equatorial plasma irregularities are different for the pre- and post midnight periods due to the ambient conditions that prevail along night.

Yizengaw et al. (2009) showed that h'F presents a peak at post-midnight hours that indicate the existence of some electrodynamic force that drives the F layer upward, creating conditions for irregularities development.

The effects of solar and geomagnetic activities on spread-F vary with latitude and longitude. Cueva et al. (2013) examined data from three equatorial stations along Solar minimum and maximum conditions. Their results showed an increase in the

spread-F occurrence rate with solar flux. Although many researchers have discussed the characteristics of spread-F irregularities at equatorial and low latitudes, some issues still needed for better understanding of their spatial and temporal variability of spread-F and plasma bubbles. So, the analysis of long term-data was performed in this work covering high and low solar activity conditions with spread F echoes observations over the Central Pacific region using the VHF radar installed in Christmas Island. In this study we present results from data analysis of echoes distribution using the 50-MHz Christmas

Island radar along 2003 and 2012 time-period. The observations allowed us to determine how the echoes vary with local time and height throughout different seasons and solar flux conditions.

## 2 Measurements and Analysis

### 2.1 VHF radar measurements

The Christmas Island VHF radar provides data of meter-scale F-region irregularities routinely, being initially operated by Stanfor Research Institute - SRI International (2002-2007) and then operated by the US Air Force Research Laboratory (AFRL). The system uses a 100 m x 100 m coaxial collinear (COCO) antenna array. Two stationary beams were used for measurements. One beam is pointed North (azimuth $0^{o}$ and elevation $84.5^{o}$), and the other one is pointed to the east (azimuth $90^{o}$ and elevation $60.5^{o}$). The coherent radar detects fluctuations related to the plasma instabilities called field-aligned irregularities, then detection of such irregularities requires the antenna to be pointing perpendicular to the geomagnetic field line (Tsunoda et al., 2000; Tsunoda et al., 1979). Then, the north beam antenna was choosen due to be pointed in the north direction to reach perpendicularly to the magnetic field line. More technical details of this radar can be found at Miller et al. (2009). Its geographic position is 2.0º N, 157.4º W, 2.9ºN dip latitude, and its magnetic inclination (declination) varied from 4.69º ($9.36^{°}$E) in 2003 to 4.61º (9.38ºE) in 2012.

It's worth mentioning that measurements available to this study covers different solar conditions when F10.7 varied from 200 SFU (high solar flux conditions) to 66 SFU (low solar flux conditions), as shown in Figure 1. Data measurements of spread F echoes to this study are between the period of January 2003 and December 2012. All our data are presented as altitude integration from 200 km to 1000 km height as function of signal to noise ratio (present in Figure 2), and the horizontal dashed lines (at 20 LT and 00 LT) representing time threshold to assist observation of time-echoes distribution. Lack of data also presented as black space in the figure, mainly for 2014 (March equinox and June solstice).

Our interest focus in the local occurrence of F-region echoes (5-meter-scale irregularities) as one of the most interesting and challenging phenomenon for space weather and climatological models. The physical mechanism responsible for this phenomenon are complexes and not fully understood. So, we organized our data attempting to present the difference in seasonal and solar flux conditions as a function of time and height of irregularities observed in the VHF-Radar. For this study, we limit our focus to quiet-time irregularities.

## 2.2 Data Analysis

It is well known that high geomagnetic activities directly cause drastic perturbations in the zonal electric field, in the equatorial and low latitude regions, affecting the growth and development of ionospheric irregularities. These perturbations can be categorized as prompt penetration (PP) and disturbance dynamo (DD) electric field (Abdu et al., 2018; Astafyeva et al., 2018; and Shreedevi & Choudhary, 2017). These perturbed electric fields occurring in the post sunset period can enhance/weaken the regular eastward electric field and vertical plasma drift, then affecting the uplift of the F layer (Fejer et al., 1991), and as a consequence affecting the generation of irregularities (Aarons. 1991; Abdu, 2012).

In sequence, to avoid the disturbed geomagnetic periods and their effects on irregularity generations, we classify the data with low geomagnetic conditions using the 3-hour Planetary K index (known as Kp). Each measurement was tagged with the value of Kp for the time, of the measurement, plus the previous 3 Kp values. We limited our study to quiet geomagnetic conditions to be those when none of the three Kp indexes exceeded 3.

The solstice is when the Sun reaches the most southerly or northerly point in the sky, while an equinox is when the Sun passes over Earth's equator. So, to sort our measurements according the four seasons Spring, Summer, Fall and Winter we use 91 days of data centered on each day 21 of March, June, September and December, respectively. We used the quiet time radar echoes for each season to obtain the occurrence rate of echoes. We establish that a good representation of irregularity occurrence is given by echoes distribution above 0 dB divided by the total number of observations. Our criterion is a good commitment among being able to identify the occurrence of spread F echoes and to eliminate the effects of non-geophysical echoes.

The sample rate of the VHF radar is estimated for every 15-minute intervals starting at 18:00 LT, right before sunset until 05:00 LT near sunrise. To construct maps of irregularity occurrence rate in function of height and local time we had computed for every 15-km height intervals starting at 200 km up to 1000 km altitude.

## 3 Results and Discussions

We can observe, in Figure 2, a significant difference in time of occurrence and duration between the spread F events at solar maximum and minimum. According with the data, during solar maximum the spread F events were observed to occur near the time when upward drift is large which is promptly after local sunset and lasting few hours, while during solar minimum the upward drift is usually short, the spread F were observed usually throughout the whole night, so upward and downward ionospheric conditions may play a role in the morphology of irregularities. Stoneback et al. (2011) showed the role of

vertical drift during the extended solar minimum and how it varies from sunset until postmidnight periods. These previous work observations increase the need for further study of climatology of echoes evolution in time and altitude.

130

In the way of understanding this climatology of spread F evolution along seasonality and solar activity we analyzed radar echoes occurrence as function of time and altitude along solar maximum and extended solar minimum periods, since the evening vertical drifts and layers heights increase noticeably with solar activity, and along nighttime.

135 **3.1 On the height variability of echo occurrence rates**

Peak altitude profiles of the occurrence rate of F-region echoes are shown in Figure 3, ~~top~~ upper panels. They were organized by seasons (March equinox, June Solstice, September equinox and December solstice, from left to right, respectively) along 2003 to 2012 period. Horizontal dashed lines placed at 250 and 350 km height to assist observation 140 (hereafter called altitude threshold).

Comparing the peak of altitude echoes along seasons, we can observe higher occurrence rates of all years over June solstice and September equinox than March equinox and December solstice seasons. This observation matches with previous result by Cueva et al. (2013), that shown the peak occurrence of equatorial spread F for this region being around July-August 145 months.

When analyzing solar minimum years (2006 and 2008) we can lay down our attention to the peak echoes altitude, which was slightly higher, in altitude, in June solstice than in September equinox. As observed in the Figure 3, we found a more prominent peak time/altitude occurrence in the September equinox (before midnight) than in the June solstice (around 150 midnight hours). For years of solar maximum (2003 and 2012) we can mention that peak altitude distribution is the highest, mainly during June and september seasons, nevertheless present minor percentage of occurrence than solar minimum years (2006 to 2008). The minimum occurrence of peak altitude occurs in March equinox which is the period of scarce spread F echoes over Christmas Island region. During solar maximum period spread F echoes have less occurrence than in solar minimum period, reaching higher altitudes as observed in June solstice 2003 when the peak altitude was higher than the 155 threshold altitude of 350 km. During September equinox higher plumes are frequently observed than in other periods which agrees with results presented by Cueva et al.(2013).

**3.2 On the time variability of echo occurrence rates**

Time variation in the occurrence rates of F-region echoes for the period in study is shown in Figure 3, lower panels, also separated by seasons (March equinox, June Solstice, September equinox and December solstice, from left to right,

respectively). The vertical dashed lines represent local sunset and local midnight. As we can observe the percentage of occurrence of echoes presents solar flux dependence. During solar maximum radar echoes are confined to a few hours after sunset, on the other hand during solar minimum echoes are more broaden out in time and can arise late in the evening after

sunset and more closely to midnight hours. As we get closer to solar minimum period the amplitude of echoes occurrence increases due to high probability to occur echoes along all night. Which is observed during years 2006 to 2008 with more amplitude than echoes observed during solar maximum, similar finding was mentioned by Niranjan et al. (2003) when analyzed spread F data from 1997-2000 period, and also by Burke et al. (2004) and Dao et al. (2011) using satellite data from different geographical locations.


Seasonal dependence of echoes along solar cycle is also observed. September equinox has more conditions to develop irregularities over the region, as explained before, as well higher echoes occurrence either for solar minimum and maximum periods. For March equinox and December solstice we have less probability of echoes occurrence as observed in the Figure 3, moreover amplitude of echoes occurrence is always lower in solar maximum than in solar minimum. Is important to

mention that for June solstice the echoes are especially observed around local midnight and post-midnight hours, which is in agreement with observations made by Otsuka et al. (2018) during solar minimum period.

Under quiet magnetically conditions, and solar minimum there are some possible seeding mechanisms competing that

increase the probability for spread F generation along all night (pre-midnight and post-midnight), as well as uplifting the F layer. For example, gravity waves, launched from active convection region in the troposphere, could propagate into the ionosphere (Takahashi et al., 2009, 2010; Maurya et al., 2020; Correia et al., 2020) and contribute to the instability seeding. Another is the Medium-scale traveling ionospheric disturbances (MSTID) activity providing perturbations in the electric fields for the low latitude F region to be unstable at postmidnight hours, that can seed the RT instability at the magnetic

equator (Otsuka et al., 2009, Yokoyama et al., 2011 and Narayanan et al., 2019). Another mechanism could be the uplift of the F layer around midnight (Nicolls et al., 2006) caused by decreasing westward electric field in conjunction with sufficient recombination and plasma flux. However, the causes of midnight F-layer increase are not yet clearly established.


For the extended solar minimum period, during June solstice and December solstice months, we observed post-midnight echoes similar as previously reported by Otsuka et al. (2012) during 2005 to 2009 period. September equinox also presents post-midnight events for solar minimum period. Our findings are summarized in Figure 4. On top panel is presented UT (LT=UT+14) in the vertical axis for the time peak echoes occurrence along solar cycle separated by seasonality. The

ionospheric sunset and the local midnight are highlighted as horizontal dashed lines, and the error bars represent the standard

deviations. We can clearly observe the peak time echoes occurrences being closer to the time of  PRE during high solar activity years, with small standard deviations, (see 2003, 2004 and 2011 and 2012) and around midnight during solar minimum conditions, with bigger standard deviations, (see years 2007 to 2009). December solstice season shows very different behavior in the years 2004 and 2012 during the high solar condition, also in 2003 the peak time echoes occurrence was very late compared with other seasons for the same year. So, further study must be necessary ~~in~~ at this point.


According to our observations, during solar minimum, the error bars (standard deviation) must be higher than during solar maximum periods due to the probability of spread F echoes occurrences, which are spread out in time (with maximum observations around local midnight) under minimum conditions and localized around to local sunset under maximum solar conditions. For example, the March equinox  and June solstice are a good representation of this behavior. They show very small deviation bars in 2003, 2011, and 2012 (solar maximum conditions) and large deviation bars from 2006 to 2009 years (solar maximum conditions). The September equinox represents the same general trend, but during 2012 the deviation bar is large, beyond the expected for solar maximum conditions.


The bottom panel in Figure 4 shows altitude peak variation along the solar cycle, also separated by seasonality. The error bars show the distribution of peak altitude echoes observations. The altitude parameter seems to follow a very good trend, being higher altitudes for solar maximum conditions and lower altitudes for solar minimum conditions. Again December solstice doesn't match very well with this trend, also presenting a lower altitude peak compared with all seasons. During 2008 the December solstice is higher than its general trend. We observe the size of error bars decreasing from solar maximum to solar minimum conditions. During solar maximum, the echoes reach higher altitudes compared with echoes occurrences during solar minimum, that's why the deviation bars are bigger during solar maximum years. The altitude parameter is an important parameter since it is one key process in the generation mechanism for ionospheric irregularities. Peak altitude echoes of June solstice reach higher altitude difference from solar maximum to solar minimum periods, when compared with March and September equinoxes which were closer to 300 km most of the solar cycle period.


## 4 Conclusions

The seasonal variability observed in the amplitude of peak echoes occurrence, either for altitude or time, is suitable for the
seasonal spread F occurrence over the Pacific region. During high solar activity spread F were observed more often after sunset and rare/uncommon observations around mignight hours. The RT instability occurs at the magnetic equator after sunset when the eastward electric fields increase and structures reaching to higher altitudes are due to vertical ExB drift at the equator, is well acknowledged for high solar flux periods. However, during the low solar cycle period observed (years 2006 to 2009) spread F did't reach higher altitudes than in high solar conditions, its appearance was very frequent around

midnight hours, and last for many hours. The mechanism that governs its appearance is no longer the prereversal enhancement because it just happen around the sunset terminator. The generation mechanism for the post-midnight irregularities at quiet time during solar minimum conditions is still not clear, or not completly understood. Some authors also found similar occurrence, in solar minimum period, of plasma density irregularities mostly after midnight (Heelis et al., 2010, Li et al., 2011  and Dao et al., 2011). So, occurrence of post-midnight events were observed to present negative

correlation with solar activity, decreasing from solar minimum to solar maximum.

So, for Christmas Island sector, we can conclude that spread F echoes occurs along all solar flux conditions. The PRE being the main mechanism for spread F generation, consequently occurrences arising closer to sunset terminator, with higher structures and short duration for solar maximum conditions. Spread F occurrence over December solstice season needs more

study since it doesn't follow the peak time occurrence for solar maximum condition. For solar minimum conditions the mechanisms necessaries for spread F generations are not clear, being  the seeding of the RT instability and the uplift of the F layer. Anyway, the spread F occurrences are happening along all night with high occurrence mainly around local midnight, with peak altitude echoes distribution remaining around 300kms, and with long time duration.

It is still not well understood what causes higher occurrences of midnight and post-midnight irregularities during the solar minimum compared with solar maximum conditions. Some theories have been raised to explain the generation mechanisms, but further investigation is needed. Studies must focus specifically on midnight and post-midnight echoes with multiple instrumentation to bring a clear understanding of the generation mechanisms.


**5 Data availability**

All raw data belong to AFRL Geospace Environment Applications and Impacts Program at Kirtland AFB. Data requirements will be made directly to AFRL directorate.


**6 Author contributions**

Ricardo Y.C. Cueva came up with the idea, prepared all data analysis, then prepared the article draft and final version. E.R. de Paula participated advicing and revieweing the manuscript. Acácio C. Neto gave support with data analysis and equipments.


## 7 Competing interest

The authors declare that they have no conflict of interest.

## 8 Acknowledgement

The authors are very grateful to R.T. Tsunoda and K.M. Groves for providing the VHF radar data from Christmas Island equatorial station, also scknowledgements for the AFRL Geospace Enviroment Applications and Impacts Program at Kirtland AFB. The author E. R. de Paula thanks the support from CNPq 302531/2019-0 as well as the INCT GNSS-NavAer grants 2014/465648/2014-2 CNPq and 2017/50115-0 FAPESP. The author R. Y. C. Cueva thanks the PIBIC/UEMA program for constant support.

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

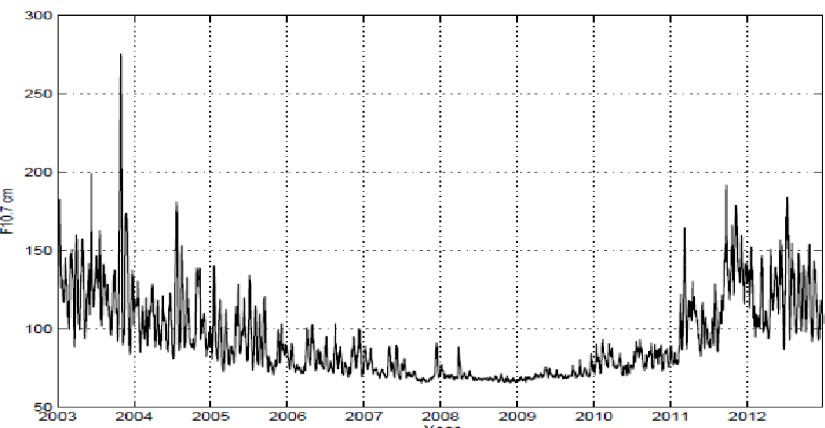

**Figure 1: Solar flux index F10.7cm covering period used in this study, which covers solar conditions where F10.7 varied from 200 SFU to 66 SFU.**




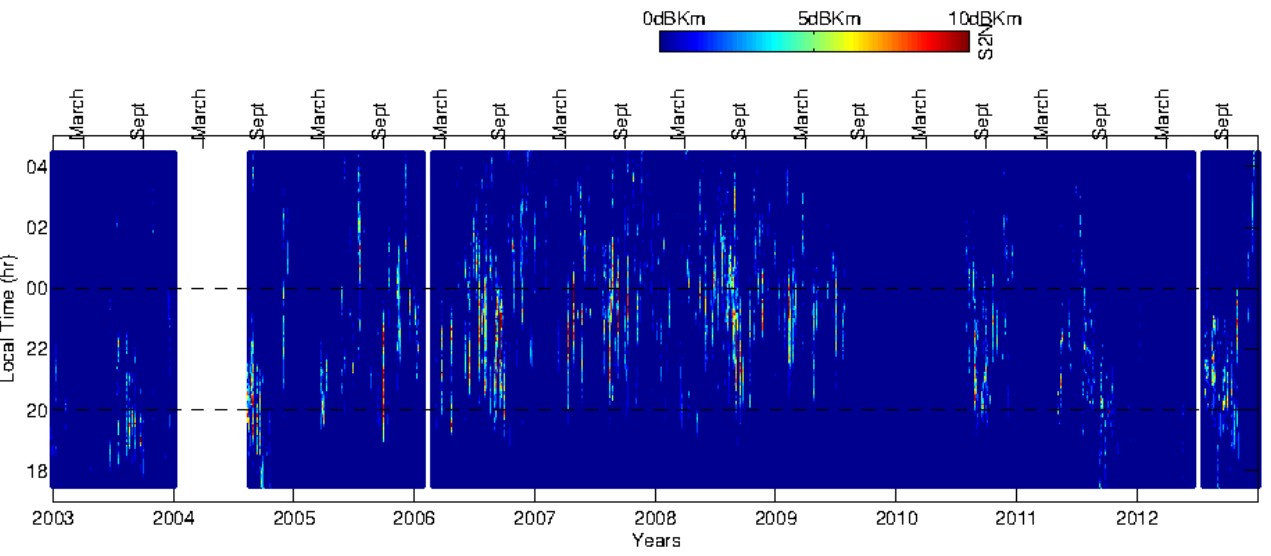


**Figure 2: VHF radar data is presented as altitude integration from 200 km to 1000 km height as function of signal to noise ratio (dBKm), the horizontal lines represent local sunset and local midnight to help observation of echoes distribution.**



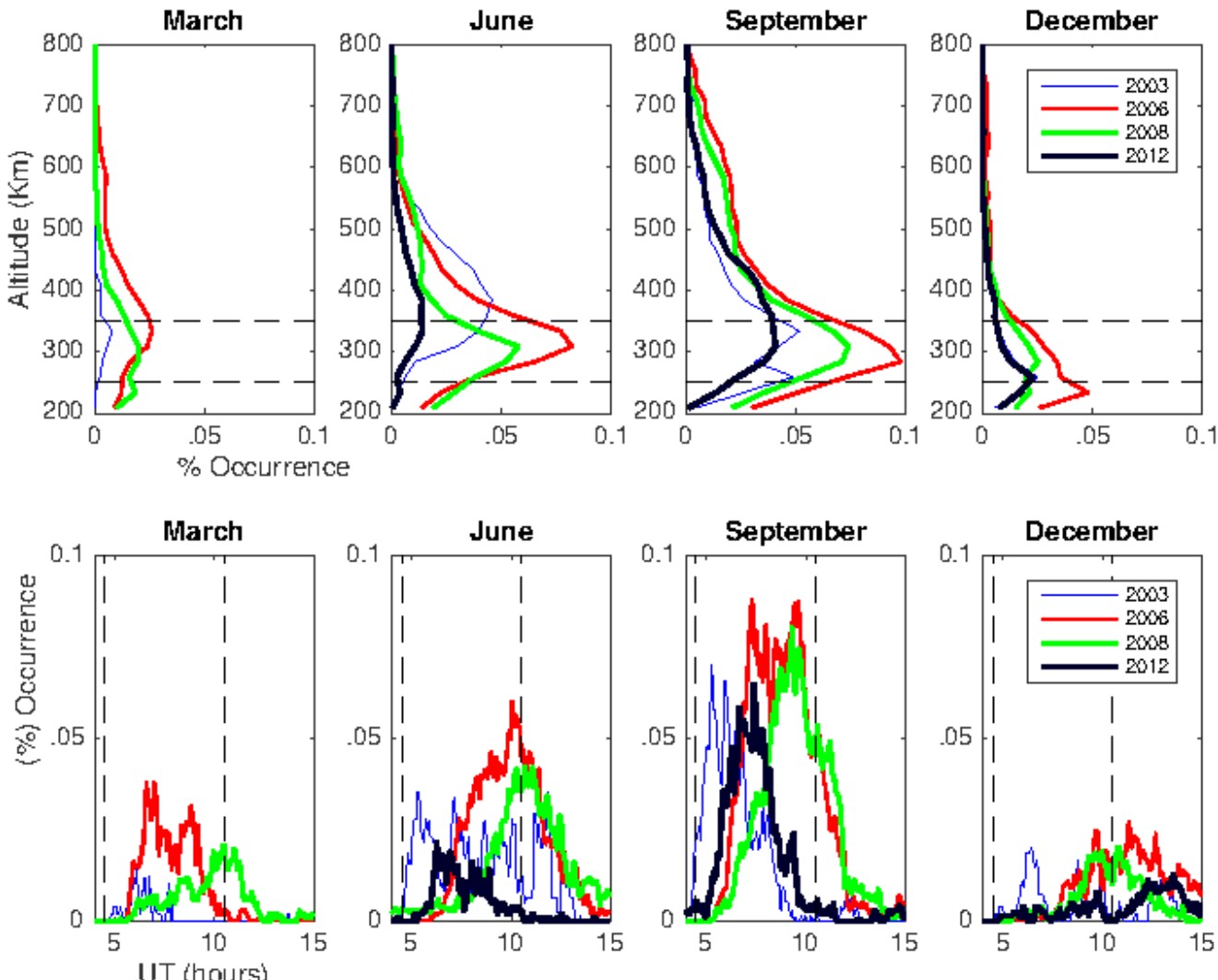


**Figure 3: Peak altitude (top panels) and time (bottom panels) variations along the years 2003, 2006, 2008 and 2012 years. Also divided by seasons.**


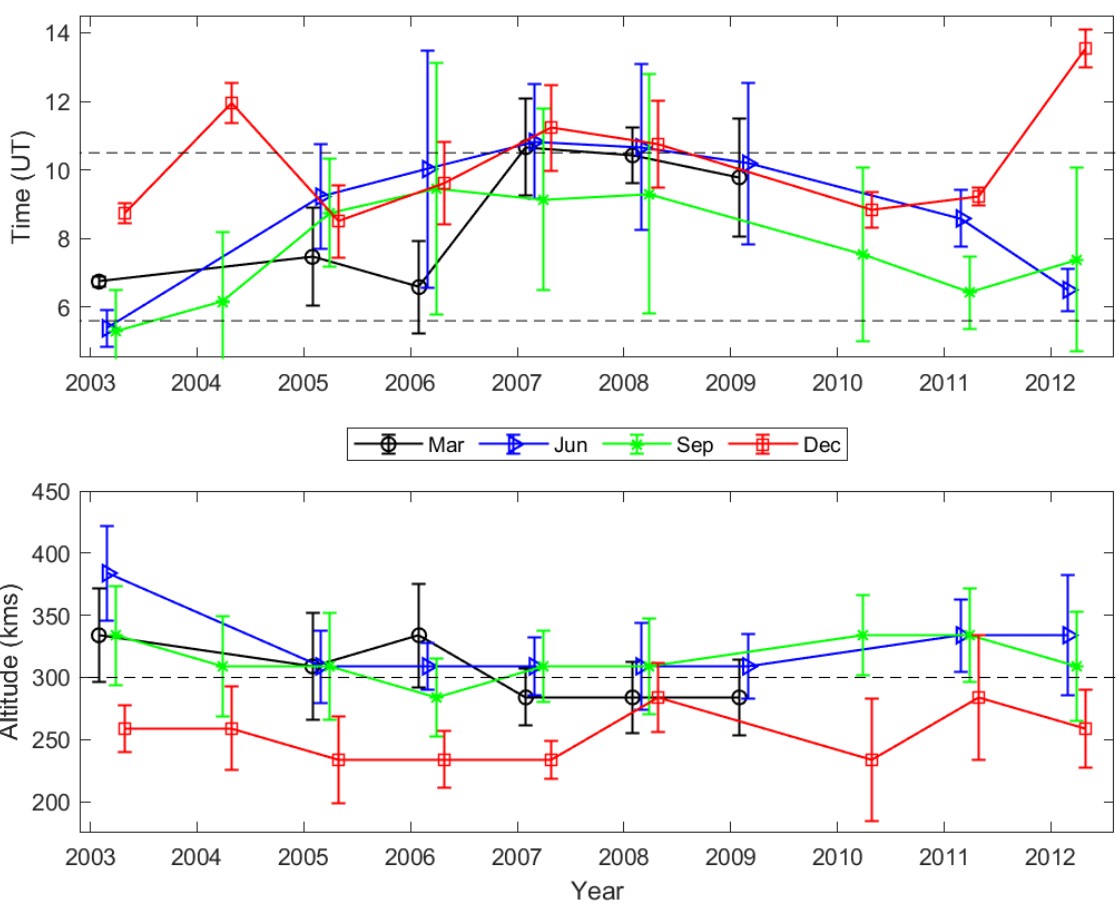

**Figure 4: Peak time (top panel) and altitude (bottom panel) echoes occurrences along solar cycle, divided by seasons. The error bars show the standard deviation of observations.**
