# Peer review of "Temporal and altitudinal variability of the Spread F observed by the VHF radar over Christmas Island"

_Annales Geophysicae, 2021_

## Author Response (AR1)

**Temporal and altitudinal variability of the spread F observed by a VHF radar over Christmas Island**

Author(s): Ricardo Yvan de La Cruz Cueva et al.
MS No.: angeo-2021-70
MS type: Regular paper
Special Issue: From the Sun to the Earth's magnetosphere–ionosphere–thermosphere

We take this opportunity to thank the editor and reviewers of our paper for their kind collaboration in the improvement of this manuscript. We have taken into account all the concerns raised and we made the suggested modifications. We have implemented numerous improvements to the paper. Below we justify our replies to the suggestions made by the respected reviewers of this paper. So, in the following, we include our answers point-by-point.

**ANSWERS TO REVIEWER 1:**

**1. The title is confusing, I guess that "Temporal and altitudinal variability of the Spread F observed by a VHF radar over Christmas Island" sounds better for the purpose of the manuscript.**

Thank you for your observation, and we agree. So, the new title:
*"Temporal and altitudinal variability of the Spread F observed by the VHF radar over Christmas Island".*

**2. Several sentences of the manuscript are not written in the usual English language, producing some misunderstanding. So, I suggest a deep revision of the concepts. A strong example is that: the work investigates Spread F from VHF radar echoes. Sometimes, the authors say that they are investigating "Spread F echoes" that is correct, but sometimes, they refer simply as echoes and it can make confusion in the reader. I suggest using the same term in all sentences or define all the terms that could have the same meaning.**

A revision of the English was done according to the referee suggestion.

**3. The definition of season presented in the lines 99-100 is not correct. It can produce different interpretations for the results. For instance, the Summer usually starts on 21 June and ends on 22 September. In my opinion, if the authors would like to emphasize the seasonal effects on the Spread F, they should use the correct definition for the season. Additionally, it can be interesting for the discussion because they will be able to make comparisons with other observations (previous works) that use the correct definition of the seasons. Only after these corrections, we can examine the real effect of the season on the Plumes over Christmas Island. I have other observations to make on this topic that are presented in the manuscript,**

**but I prefer to see whether the correct definition of the season will not address those points.**

Thank you the reviewer for the observations.

Dear reviewer, we believe that when you mention June 21 (summer) you refer to Midsummer's Day, which is based around the summer solstice, the longest day of the year., and the solstice fall in the middle of summer.

Is possible to find the same seasonal periods in papers like Koustov et al., 2019 (https://doi.org/10.1186/s40623-019-1092-9), Denardini et al., 2005 (10.1016/j.jastp.2005.04.008), and Niranjan et al., 2003.

We updated the text to:

*"The solstice is when the Sun reaches the most southerly or northerly point in the sky, while an equinox is when the Sun passes over Earth's equator. For example, June solstice, or June 21, is the longest day of the year in the northern hemisphere. So, to sort our measurements according to the four seasons Spring, Summer, Fall, and Winter we use 91 days of data centered on each day 21 of March, June, September, and December, respectively".*

**4. The authors are suggesting that the occurrence of Spread F are inversely proportional to the solar cycles. They must explore this result more and try to explain how it can be explained physically.**

The result that comes up with this study shows spread F occurring along the entire solar cycle, and showing a negative correlation with solar activity. Physically, there are two conditions for spread F occurrence, one is the seeding of the RT instability, and the other is the uplift of the F layer.

We clarify on the text as:

*"During high solar cycle spread F occurs more often after sunset and rare/uncommon observations around midnight hours. These structures reach higher altitudes. The RT instability occurs at the magnetic equator after sunset when the eastward electric fields increase and structures reaching to higher altitudes are due to vertical ExB drift at the equator, which is well acknowledged for high solar flux periods. However, during low solar cycle (years 2006 to 2009) spread F don't reach higher altitudes -as before, their appearance is very frequent around midnight hours, and last for many hours. The mechanism that governs its appearance is not longer the prereversal enhancement because it just happens around the sunset terminator. The generation mechanism for the post-midnight irregularities at a quiet time during solar minimum conditions is still not clear, or not completely understood. Some authors also found occurrence, in the solar minimum period, of plasma density irregularities mostly after midnight (Heelis et al., 2010, Li et al., 2011 and Dao et al., 2011).*

*Under quiet magnetically conditions, and solar minimum conditions there are some possible seeding mechanisms competing that increase the probability for spread F generation along all night (pre-midnight and post-midnight), as well as uplifting the F layer. For example, gravity waves, launched from the active convection region in the*

*troposphere, could propagate into the ionosphere (Takahashi et al., 2009, 2010) and contribute to the instability seeding. Another is the Medium-scale traveling ionospheric disturbances (MSTID) activity providing perturbations in the electric fields for the low latitude F region to be unstable at postmidnight hours, which can seed the RT instability at the magnetic equator (Otsuka et al., 2009, Yokoyama et al., 2011 and Narayanan et al., 2019). Another mechanism could be the uplift of the F layer around midnight (Nicolls et al., 2006) caused by decreasing westward electric field in conjunction with sufficient recombination and plasma flux. However, the causes of midnight F-layer increase are not yet clearly established".*

**5. Is Figure 4 really necessary in the Conclusion section? Why do not the author include it in the result and Result and Discussion section to explore better the results?**

Thanks for the comment, we just re-wrote the text about Figure 4 as follows:

*"Our findings are summarized in Figure 4. On the top panel is presented UT (LT=UT+14) in the vertical axis for the time peak echoes occurrence along the solar cycle separated by seasonality. We can clearly observe the peak time echoes occurrences being closer to the time of PRE during high solar activity years (see 2003, 2004 and 2011, and 2012) and around midnight during solar minimum conditions (see years 2007 to 2009). December solstice season during high solar conditions is not following this trend, and further study must be necessary at this point.*

*The bottom panel in Figure 4 shows altitude peak variation along the solar cycle, also separated by seasonality. The altitude parameter seems to follow a very good trend, being higher altitudes for solar maximum conditions and lower altitudes for solar minimum conditions. Again December solstice doesn't match very well with this trend. The altitude parameter is an important parameter since it is one key process in the generation mechanism for ionospheric irregularities. Peak altitude echoes of June solstice reaches higher altitude difference from solar maximum to solar minimum periods, when compared with March and September equinoxes which were closer to 300kms most of the solar cycle period".*

**6. Line 59: "large data" => "long term data".**

Thak the referee #1 for this observation, so, we agreed and made the suggested correction.

**7. What is "SRI" in line 71?**

The acronym SRI stands for Standford Research Institute. The SRI/Geospace Division supported the VHF radar from 2002 to 2007, under the coordination of Dr. R. Tsunoda, with National Science Foundation's grants.

The text added:
*"Stanford Research Institute – SRI International."*

**8. Line 74: Please, explain what is the reason to use the North beam of the radar only.**

We apologize for not being clear on this point, we added a text in section 2.1 VHF radar measurements:

*"The coherent radar detects fluctuations related to the plasma instabilities called field-aligned irregularities, then detection of such irregularities requires the antenna to be pointing perpendicular to the geomagnetic field line (Tsunoda et al., 2000; Tsunoda et al., 1979). Then the north beam antenna was chosen due to being pointed in the north direction to reach perpendicularly to the magnetic field line".*

**9. Lines 85-89: I guess it can be removed to another section. It is not necessary in the Data analysis description**

We thank the referee for this comment. We moved this paragraph to section 2.1 VHF radar measurements.

**10. Lines 91-94: The authors must remind that the disburbed dynamo is another phenomenon that can produce unexpected behaviour in the dynamics of the F region at low latitudes in addition to the prompt penetration electric field..**

Thank you, is good to mention.

We organize the text in the manuscript as follows:

*"It is well known that high geomagnetic activities directly cause drastic perturbations in the zonal electric field, in the equatorial and low latitude regions, affecting the growth and development of ionospheric irregularities. These perturbations can be categorized as prompt penetration (PP) and disturbance dynamo (DD) electric field (Abdu et al., 2018; Astafyeva et al., 2018; and Shreedevi & Choudhary, 2017). These perturbed electric fields occurring in the post-sunset period can enhance/weaken the regular eastward vertical plasma drift, then affecting the uplift of the F layer (Fejer et al., 1991), and as a consequence affecting the generation of irregularities (Aarons. 1991; Abdu, 2012).*

*In sequence, to avoid the disturbed geomagnetic periods and their effects on irregularity generations, we classify ..."*

**11. Lines 112-116: This paragraph could be shifted to a place after the presentation of the results. It could help the author in the discussion.**

Sure, we agree with your comment dear referee. So, we moved this paragraph to the beginning of section 3.2.

**12. Line 131: "... provided by Digisondes." I could not see those profiles in the chart of Figure 3 and 4.**

By curiosity, we were trying to compare the profile of the figure with the usual digissonde density profiles from other stations like Sao Luis (Brazil). We removed this from the manuscript.

**13. Lines 174-177: I do not agree with the author that it is clear in Figure 4.**

We clarify the explanation as:

*"Our findings are summarized in Figure 4. On the top panel is presented UT (LT=UT+14) in the vertical axis for the time peak echoes occurrence along the solar cycle separated by seasonality. We can clearly observe the peak time echoes occurrences being closer to the time of PRE during high solar activity years (see 2003, 2004 and 2011 and 2012) and around midnight during solar minimum conditions (see years 2007 to 2009). December solstice season during high solar conditions is not following this trend, and further study must be necessary at this point".*

**14. Lines 179-181: I guess these conclusions are not totally supported by the results. However, after the revision of the seasons, the authors can do a check.**

We re-wrote the sentence by:

*"So, for the Christmas Island sector, we can conclude that spread F echoes occurs along with all solar flux conditions. The PRE is the main mechanism for spread F generation, consequently, occurrences arising closer to the sunset terminator, with higher structures and short duration for solar maximum conditions. Spread F occurrence over the December solstice season needs more study since it doesn't follow the peak time occurrence for solar maximum condition. For solar minimum conditions, the mechanisms necessary for spread F generations are not clear, being the seeding of the RT instability and the uplift of the F layer. Anyway, the spread F occurrences are happening along all night with high occurrence mainly around local midnight, with peak altitude echoes distribution remaining around 300kms, and with long time duration".*

We take this opportunity to thank the editor and reviewers of our paper for their kind collaboration in the improvement of this manuscript. We have taken into account all the concerns raised and we made the suggested modifications. We have implemented numerous improvements to the paper. Below we justify our replies to the suggestions made by the respected reviewers of this paper. So, in the following we include our anwers point-by-point.

**ANSWERS TO REVIEWER 2:**

**Line 69: sub-section title of 2.1: "Data measurements" to be revised as "VHF radar measurements".**

We thank the referee for this comment. We made the suggested correction.

**Line 83, "Mach": correct to "March"**

We thank the referee for this comment. We made the suggested correction.

**Line 87, "we had organized": correct to "we organized"**

We thank the referee for this comment. We made the suggested correction.

**Line 91, "Is": correct to "It is"**

We thank the referee for this comment. We made the suggested correction.

**Line 131, "The higher occurrence of echoes in altitude is compared with the density profiles provided by Digisondes": Please show the digisonde data for comparison.**

By curiosity we were trying to compare the profile of the figure with the usual digissonde density profiles from other stations like Sao Luis (Brazil). We removed this from the manuscript.

**Line 133, "June equinox": correct to "June solstice". In the next pages there are several phrases of "March solstice, September solstice" please correct them to March equinox and September equinox.**

We thank the referee for this comment. We made the suggested correction.

**Line 134 "even when its occurrence was the opposite": what does it mean ?**

Sorry for not being clear, but I meant that the Peak echoes altitude for solar minimum conditions, was slightly higher in altitude in June solstice than September equinox,

however higher occurrence rates were higher in September equinox than in June solstice, also for solar minimum conditions.

We change the text as follows:

*"When analyzing solar minimum years (2006 and 2008) we can lay down our attention to the peak echoes altitude, it was slightly higher, in altitude, in June solstice than in September equinox. For the occurrence rates of peak time we observed being bigger in September equinox, and peak altitude occurrence before midnight (as in bottom panel), than in June solstice with peak altitude occurrence around midnight hours (as in bottom panel)".*

**Line 139, "The altitude distribution of echoes above 350 km also presents same behavior as below this threshold": what threshold ??**

We re-wrote the sentence by:

*"… Horizontal dashed lines were placed at 250 and 350 km height to assist observation (hereafter called altitude threshold)."*

*"During solar maximum period spread F echoes have less occurrence than in solar minimum period, reaching higher altitudes as observed in June solstice 2003 when the peak altitude was higher than the threshold altitude of 350 km".*

**Line 156, "September solstice": to be "September equinox"**

We thank the referee for this comment. We made the suggested correction.

**Line 315, Figure 2: If the authors plot ionospheric sunset hours in the figure, it would be useful.**

Dear referee, plotting the local sunset and the ionospheric sunset can pollute the Figure since the difference is small, around 1.1 hours. So, we plotted the ionospheric sunset in Figure 4.

**Line 326, Figure 4: please plot STD error bar for each plot, so that readers could evaluate the difference between them.**

Thanks to the referee for his observation. We prepared the Figure and add an explanation to the text.

[Figure]

*Figure 4: Peak time (top panel) and altitude (bottom panel) echoes occurrences along solar cycle, divided by seasons. The error bars show the standard deviation of observations.*

---

## Author Response (AR2)

Temporal and altitudinal variability of the spread F observed by a VHF
radar over Christmas Island
Author(s): Ricardo Yvan de La Cruz Cueva et al.
MS No.: angeo-2021-70
MS type: Regular paper
Special Issue: From the Sun to the Earth's magnetosphere–ionosphere–thermosphere

We thank the editor and reviewers of our paper for their kind collaboration in the improvement of this manuscript. We have taken into account all the minor comments raised and we made the suggested modifications. Below we justify our replies to the suggestions made by the respected reviewers of this paper. So, in the following, we include our answers point-by-point.

**Answers to reviewer 01:**

Second review to the manuscript "Temporal and altitudinal variability of the Spread F observed by the VHF radar over Christmas Island"

General comment:

The authors attended the minor comments I noted, but did not reply on my general comment.
"The data and the statistical analyses are interesting and worth to publish. However, the authors did not try to explain, quantitatively or qualitatively, why it occurred".

If this present manuscript is to be considered as a full research paper, the discussion of the physical explanation of what they observed will be necessary.

The Chapter 4 (Conclusion) is necessary to revise, making to be concise.

We had reorganized chapter 4.

Language corrections are necessary before to be accepted.
My review conclusion: the present manuscript is necessary to have a major revision and language editing.

Minor comment are follows:

Line 13, "Raileight" : correct to "Rayleigh"
Line 37, "recently": newly
Line 58, "low solar conditions": low solar activity conditions.
Line 102," eastward vertical plasma drift": eastward electric field and vertical plasma drift ??
Lines 109-110, "The solstice is when the Sun reaches the most southerly or northerly point in the sky, while an equinox is when the Sun passes over Earth's equator. For example, June solstice, or June 21, is the longest day of the year in the northern hemisphere. ": This sentence is obviously not necessary, I guess.

*Thanks to the reviewer for his observations, só, we agreed and made the suggested corrections.*

Line 120: I could not find any explanation of Figure 2, which is the main observational result of the present paper.

*We had highlighted the explanation for Figure 2 as:*
*"We can observe, in Figure 2, a significant difference in time of occurrence and duration between the spread F events at solar maximum and minimum. According with the data during solar maximum the spread F events were observed to occur near the time when upward drift is large which is promptly after local sunset and lasting few hours, while during solar minimum the upward drift is usually short, the spread F were observed usually throughout the whole night...."*

Line 136, "top panel": upper panel,

*Thanks to the reviewer for his observations, só, we agreed and made the suggested corrections.*

Line 148 "observed being bigger": please revise this English phrase.

*After correcting we end up with: "As observed in the Figure 3, we found a more prominent peak time/altitude occurrence in the September equinox (before midnight) than in the June solstice (around midnight hours)."*

Line 177 "The variability over seasonality": seasonal variability ?

*Thanks to the reviewer for his observations, só, we agreed and made the suggested minor comment.*

Line 203-205, "We can clearly observe the peak time echoes occurrences being closer to the time of PRE during high solar activity years (see 2003, 2004 and 2011 and 2012) and around midnight during solar minimum conditions (see years 2007 to 2009).": this is what the authors found in the data analysis.

*Yes, and we move this paragraph to the Discussion section.*

Line 175 "Conclusion": This chapter has 6 paragraphs, too long. Some parts are to be in the "Discussion". Only two paragraphs, Lines 203-205 and lines 215-217, are mentioning some finding.

*We re-organize the discussion and conclusion paragraphs.*

**Answers to reviewer 02:**

The authors addressed my concerns and the manuscript was improved. In my opinion, it can be published after technical correction.

The text written from the lines 188 to 220 seems to be part of the discussion and not conclusions.

*We re-organize the discussion and conclusion paragraphs.*